# Differential Metabolites in Osteoarthritis: A Systematic Review and Meta-Analysis

**DOI:** 10.3390/nu15194191

**Published:** 2023-09-28

**Authors:** Zeqi Liao, Xu Han, Yuhe Wang, Jingru Shi, Yuanyue Zhang, Hongyan Zhao, Lei Zhang, Miao Jiang, Meijie Liu

**Affiliations:** 1Medical Experimental Center, China Academy of Chinese Medical Sciences, Beijing 100700, China; 13611244269@163.com (Z.L.); 18801094991@163.com (Y.W.); shijingru2023@163.com (J.S.); zhangyuanyue0306@163.com (Y.Z.); zhaohongyan@merc.ac.cn (H.Z.); 2Institute of Basic Research in Clinical Medicine, China Academy of Chinese Medical Sciences, Beijing 100700, China; zyrhanxu@163.com; 3National Data Center of Traditional Chinese Medicine, China Academy of Chinese Medical Sciences, Beijing 100700, China; leizhang@ndctcm.cn

**Keywords:** osteoarthritis, differential metabolites, metabolomics, meta-analysis

## Abstract

(1) Many studies have attempted to utilize metabolomic approaches to explore potential biomarkers for the early detection of osteoarthritis (OA), but consistent and high-level evidence is still lacking. In this study, we performed a systematic review and meta-analysis of differential small molecule metabolites between OA patients and healthy individuals to screen promising candidates from a large number of samples with the aim of informing future prospective studies. (2) Methods: We searched the EMBASE, the Cochrane Library, PubMed, Web of Science, Wan Fang Data, VIP Date, and CNKI up to 11 August 2022, and selected relevant records based on inclusion criteria. The risk of bias was assessed using the Newcastle–Ottawa quality assessment scale. We performed qualitative synthesis by counting the frequencies of changing directions and conducted meta-analyses using the random effects model and the fixed-effects model to calculate the mean difference and 95% confidence interval. (3) Results: A total of 3798 records were identified and 13 studies with 495 participants were included. In the 13 studies, 132 kinds of small molecule differential metabolites were extracted, 58 increased, 57 decreased and 17 had direction conflicts. Among them, 37 metabolites appeared more than twice. The results of meta-analyses among four studies showed that three metabolites increased, and eight metabolites decreased compared to healthy controls (HC). (4) Conclusions: The main differential metabolites between OA and healthy subjects were amino acids (AAs) and their derivatives, including tryptophan, lysine, leucine, proline, phenylalanine, glutamine, dimethylglycine, citrulline, asparagine, acetylcarnitine and creatinine (muscle metabolic products), which could be potential biomarkers for predicting OA.

## 1. Introduction

Osteoarthritis (OA) is a progressive disorder affecting mobile joints [1]. It is characterized by cartilage degradation, bone remodeling, fragmentation and inflammation, resulting in joint pain, swelling and stiffness that lead to limited mobility [2]. OA can also cause sleep disturbances, fatigue and feelings of depression or anxiety; in particular, the constant pain greatly affects physical and mental functioning, which significantly impacts an individual’s quality of life [3,4,5]. According to the recent 2017 Global Burden of Disease Study, the prevalence of OA currently exceeds 300 million, with an incidence of nearly 15 million [6]. It is estimated that the prevalence of OA will continue to accelerate in the coming years due to the increased prevalence of obesity and the global increase in life expectancy [7]. In 2019, it was estimated that OA had contributed approximately USD 460 billion in all-cause medical costs in the United States [8].

The pathological process of OA involves a complex interplay of immune, metabolic, hormonal, and genetic factors, leading to the damage and destruction of joint cartilage, bone, synovium, and other tissues, which can result in joint pain and functional impairment [9,10,11,12]. However, the exact mechanisms that underlie this disease remain unclear [13]. Therefore, until now, no effective therapeutic intervention capable of modifying the pathological progression of OA has been identified, and available treatment options are limited to pharmacological and non-pharmacological measures, with surgery as the final option [14,15,16]. Nonetheless, the long-term consumption of analgesics and anti-inflammatory drugs has been associated with a range of side effects, which include gastrointestinal, cardiovascular, and hepatorenal toxicity [17]. Despite the substantial benefits of joint replacement surgery including pain relief and improved function, the procedure is cost-intensive and associated with high surgical risk, and requires a substantial post-operative recovery period and follow-up care [18,19].

Meanwhile, several studies have reported that early intervention strategies such as exercise therapy and joint manipulation can alleviate symptoms and reduce the severity of OA [20]. Thus, preventative measures and early intervention in the management of OA have been emphasized. Nevertheless, current diagnostic techniques such as imaging and joint symptoms may not enable an accurate prognosis of the disease [21], and so superior techniques for early diagnostic and preventative strategies for OA are much-needed.

Currently, biomarkers have emerged as a promising objective diagnostic tool for predicting disease progression, determining its early stages, and guiding therapeutic interventions [22]. Detecting biomarkers in bodily fluids such as urine or blood enables the timely detection of potential abnormalities, thus reducing unnecessary medical procedures and hospital visits. Early detection and diagnosis based on biomarkers allows the implementation of tailor-made preventative care plans, mitigating future healthcare costs and providing more comprehensive care [23]. Besides this, the discovery of early biomarkers of the disease might also reveal the onset and progression mechanisms of OA from a metabolic perspective [24].

The development of metabolomics technologies has facilitated the detection of early biological alterations in OA through changes in metabolites [25]. These technologies also enable the comparison of metabolomic differences between healthy individuals and those with OA, leading to the identification of disease-associated biomarkers and further investigations of their underlying mechanisms [26]. This approach offers deeper insights into the disease’s pathogenesis, and opportunities for early treatment that may advance patient outcomes and consequently increase their quality of life [27]. Recent studies have demonstrated the efficacy of metabolomics techniques based on mass spectrometry platforms in accurately detecting and identifying small molecule metabolites [28]. Notably, metabolomics based on mass spectrometry shows great potential as a promising biomarker and drug discovery tool for multiple sclerosis, while also advancing our understanding of the gut microbiota [29,30]. Therefore, the identification of biomarkers is of paramount importance for the early diagnosis, treatment, and prevention of OA, potentially alleviating the burden on individuals and society as a whole.

An increasing number of people are studying OA metabolomics [31,32], yet there are still no consistent conclusions about the differences in metabolites between OA patients and healthy individuals that have been yielded through comprehensive literature reviews. Clear knowledge related to this important issue would definitely facilitate a better understanding and measurement of this disease. Therefore, we conducted a systematic review and meta-analysis of small molecule differential metabolites between OA patients and healthy individuals to provide a useful basis for future metabolomic studies of OA samples.

## 2. Materials and Methods

The systematic review and meta-analysis was undertaken via the Preferred Reporting Items for Systematic Reviews and Meta-Analyses Protocol (PRISMA-P) [33]. This study has been registered in the International Prospective Register of Systematic Reviews (PROSPERO CRD42022365740).

### 2.1. Selection Criteria

Inclusion criteria: (1) patients with a clear diagnosis of OA; (2) metabolomics technology was used to measure and analyze blood, urine and joint synovial fluid samples from patients and healthy controls (HC); (3) human-based randomized controlled trials, cohort studies, and case–control studies; (4) reports on small molecule differential metabolites between OA patients and healthy individuals. Exclusion criteria: (1) duplicate publication; (2) the full text is not available; (3) non-primary documents (reviews, commentaries, editorials, or letters); (4) insufficient information.

### 2.2. Search Strategy

All the records were identified through the EMBASE, the Cochrane Library, PubMed, Web of Science, WanFang Data, VIP and CNKI up to 11 August 2022, with the keywords and synonyms referring to OA and metabolomics (Appendix A). No restrictions were placed on the date and language of publication. In addition to the electronic database search, a thorough manual search of the reference lists of all selected articles was undertaken. This was done to ensure that no relevant studies were inadvertently missed during the initial search process. The screening of bibliographies was conducted independently to further strengthen the comprehensiveness of the search.

### 2.3. Study Selection

All of the identified records were downloaded to Endnote X9, then the duplicates were removed. Two independent researchers (ZQL and YHW) respectively screened the studies by title and abstract, then the studies meeting our criteria were found, and their full texts were taken for further screening. Any disagreements were discussed with the third researcher (MJL) until the team reached a consensus.

### 2.4. Risk of Bias Assessment

Two independent researchers (ZQL and YHW) assessed the risk of bias using the NOS [34], which is a tool for assessing the quality of an observational study. The studies were evaluated by a “star system” based on three aspects, including selection, comparability and exposure. Over 7 stars was considered as high quality. Inversely, studies not reaching 7 stars were removed from the sensitivity meta-analysis.

### 2.5. Data Collection

The necessary information was extracted from each study, including the following items: the name of the first author; year of publication; country of origin; type of study design; language; sample size; age; diagnostic criteria of patient; whether an independent validation queue was used; metabolomics techniques and trends; concentrations of small molecule differential metabolites in OA patients and HC. The data were derived from the article and the article’s additional files. When the data were only shown in graphs and details could not be obtained from original authors, we used WebPlotDigitizer (Web Plot Digitizer, V.4 6, San Francisco, CA, USA: Ankit Rohatgi, 2022) to extract data from graphs.

To convert the median and quartile to mean and standard deviation (SD), we first tested the skewness and then applied a new piecewise function based on the sample size [35,36]. When two experimental groups were reported but matched with one control group, the mean and standard deviation of the experimental groups were merged using the formula provided by the Cochrane Manual (Appendix A).

### 2.6. Data Synthesis

Qualitative analysis was performed for summarizing the changing directions of differential small molecule metabolites in OA by counting the frequency across the studies. Then, the concentrations of small molecule metabolites were summed up via a meta-analysis across studies using the MD and 95% CI. The studies’ heterogeneity values were measured by the *I*^2^ statistic; *I*^2^ > 50% indicated the presence of significant heterogeneity, and a random-effects model would be selected; *I*^2^ ≤ 50% referred to a fixed-effects model. *I*^2^ >60% represented substantial heterogeneity. Sensitivity analyses were performed by removing high-risk studies to investigate the influence of possible bias, and by the sequential omission of individual studies. Funnel plots and the Egger’s test were used to assess publication bias when feasible (10 or more studies) [37]. All the data synthesis was performed by the Review Manager (Version 5.4.1).

## 3. Results

### 3.1. Literature Search

A total of 3798 publications were identified from the databases, and 2547 articles remained after the removal of duplicate studies; 2503 publications were further excluded after evaluating their titles and abstracts. Of the remaining 44 studies, 1 was excluded due to incomplete information, 3 were excluded due to no access to the full text, and 27 were not eligible according to our criteria. At last, 13 studies [38,39,40,41,42,43,44,45,46,47,48,49,50] were included for qualitative synthesis. Of those 13 studies, 4 studies [38,44,47,48] with 495 participant (242 in OA group, 253 in healthy control group) were recruited into the meta-analysis. The flow chart is shown in Figure 1.

### 3.2. Characteristics of Included Studies

All of the included studies were published between 2009 and 2022 and designed as case–control studies. Eight studies [43,44,45,46,47,48,49,50] were reported in English and five studies [38,39,40,41,42] in Chinese. The participants were recruited from Spain [46], the UK [43], Estonia [47], Canada [44] and China [38,39,40,41,42,45,48,49,50]. All studies were well-matched by age and sex, except one study [45] that failed to report the age information. Differential small molecule metabolites were mostly analyzed using Gas Chromatography–Mass Spectrometry (GC-MS) [40,41,42,50]. The characteristics of each study are shown in Table 1.

A total of 132 small molecule metabolites with differential expression were extracted from 13 studies, out of which 37 metabolites occurred more than twice and could be qualitatively synthesized. A total of 39 differential small molecule metabolites were extracted from four studies, of which 17 were reported with accurate concentrations in at least two studies, which enabled a meta-analysis. In the four studies, the concentration information was directly obtained in three articles [38,47,48] and extracted from the box plots by software in one [44]. These metabolites were measured in blood samples, with a concentration unit of μmol/L. The specific characteristics of these differential small molecule metabolites can be checked in Appendix A.

### 3.3. Risk of Bias of Included Studies

Based on the NOS criteria, the maximum score of the included studies in this meta-analysis, awarded on the basis of eight risk assessment items, was 9 stars. Studies with a score of 5 stars or higher were regarded as of medium-to-high quality; otherwise, they were categorized as poor quality and excluded. The studies included in this analysis exhibited a generally high level of quality, as 11 out of 13 achieved at least 7 stars [38,39,40,41,43,45,46,47,48,49,50]. One study [42] received 6 stars due to a low exposure score, and another [44] received only 5 stars due to a failure to control for a potentially important factor. This implied that all the included studies were of medium-to-high quality, and the data extracted were eligible for the meta-analysis. The results of risk of bias for the included studies are shown in Table 1. The details of the assessments for every item are listed in Appendix A.

### 3.4. Qualitative Synthesis

In the 13 studies, 132 differential small molecule metabolites were qualitatively synthesized by counting the frequency of change direction; compared with HC, 58 increased, 57 decreased, and 17 showed direction conflicts. Among them, 37 metabolites appeared more than twice. There were 11 increased metabolites, with leucine being the most prominent. On the other hand, nine metabolites showed a decrease, with citrulline, creatinine, and phenylalanine exhibiting the largest declines. Among the 17 metabolites with direction conflicts, glycine decreased 5-fold and glutamine 4-fold, while histidine, proline and threonine decreased 3-fold; alanine increased 4-fold. According to the classification by sample type, there were 33 metabolites showing increases, 41 showing decreases, and 16 showing inconsistent trends in the blood samples. In the urine samples, 22 metabolites increased, 18 decreased, and 7 showed inconsistent trends. In the synovial fluid samples of knee joints, five metabolites increased and two showed inconsistent trends. The changes in the metabolites in the blood, urine, and synovial fluid samples of patients with osteoarthritis and healthy individuals are shown in Table 2. The qualitative analysis results of 37 differential small molecule metabolites are shown in Figure 2. Details of 132 small molecule differential metabolites can be found in Appendix A.

### 3.5. Meta-Analysis

A meta-analysis of 17 differential small molecule metabolites in four studies was conducted, among which 3 were increased by concentration (tryptophan, lysine and leucine) and 8 decreased (proline, phenylalanine, glutamine, dimethylglycine, creatinine, citrulline, asparagine and acetylcarnitine), while no statistically significant changes were observed in the levels of valine, tyrosine, serine, arginine, alanine and 4-Hydroxy-L-proline. The results are shown in Figure 3 and Table 3.

Although substantial heterogeneity was detected in these comparisons, the direction of the effect estimates was relatively consistent across studies. Among the meta-analysis of 17 small molecule metabolites, only leucine [38,44,47,48] and citrulline [38,44,48] (having at least two concentrations) could be analyzed for sensitivity. The sensitivity analysis was performed by removing studies with a high risk of bias. Citrulline had significantly lower heterogeneity after the exclusion of one study [44], possibly because the data from this study were extracted by box plot and there was some error in the data. However, both leucine and citrulline did not show obvious and significant changes compared with the previous results. In addition, unfortunately, we were unable to assess publication bias using funnel plots and Egger’s test because the numbers of included studies were all less than 10.

## 4. Discussion

Overall, our meta-analysis identified 11 differential small molecule metabolites, mainly amino acids (AAs) and their derivatives or their metabolites, with 3 metabolites increased and 8 decreased in the sera of patients with OA compared to healthy subjects. Although there was some heterogeneity in the results of the meta-analysis, the direction of effect estimation was relatively consistent across studies. These metabolites may be potential candidate biomarkers and deserve further exploration.

OA is a degenerative joint disease characterized by low-grade inflammation and clinical heterogeneity [51]. Abnormal metabolic activity in chondrocytes is a reaction to changes in the inflammatory microenvironment [52]. In healthy joints, chondrocytes maintain a physiological and metabolic equilibrium, displaying potent oxidative and reduction control in the differentiation and generation of cartilage [53,54]. However, under the conditions of joint inflammation and the OA environment, chondrocytes experience pathological changes in metabolic balance and cartilage remodeling, resulting in cellular aging processes such as enhanced glycolytic pathways, mitochondrial dysfunction, and cartilage aging [10,55]. Under environmental stress, chondrocytes tend to adapt to changes in the microenvironment by switching from one metabolic pathway to another, which is associated with changes in mitochondrial dysfunction, enhanced anaerobic glycolysis, lipid, and amino acid metabolism [56].

AAs are small molecules that play a crucial role in hormone and low-molecular-weight biologically active substances, intimately linked with essential biological functions [57,58]. Emerging evidence suggests that alterations in amino acid metabolism might contribute to changes in inflammation and oxidative stress dynamics [59,60]. Dysregulating the metabolism of AAs can lead to the increased production of inflammatory mediators and reactive oxygen species, causing damage to and death of chondrocytes [61]. Evidence suggests that inflammation is a major driver of OA, as it leads to the release of matrix-degrading enzymes and alters metabolic balance by increasing the production of cytokines, chemokines, and other immune modulators, thus favoring joint destruction [62,63,64].

Tryptophan, lysine, and leucine are essential AAs and important components of proteins [65,66,67]. There is a relationship between their abnormal metabolism and the development and progression of OA [68,69]. First, abnormalities in the tryptophan metabolic pathway may lead to a shift of tryptophan to the synthetic pathway of inflammatory mediators, while reducing the utilization of tryptophan to the synthetic pathway in the growth and repair process, which may lead to damage of the osteoarticular cartilage [70,71]. In addition, tryptophan can be metabolized into serotonin, which has important effects on the growth and differentiation of articular cartilage cells [72], and a deficiency of serotonin may lead to articular cartilage damage [73,74]. Second, lysine and leucine are also associated with OA. Lysine and leucine deficiency may lead to poor cartilage and bone development, thus increasing the risk of OA [75,76]. In addition, metabolites of both lysine and leucine can have effects on bone health; for example, β-hydroxy-β-methyl butyric acid (HMB), which is produced by the metabolism of leucine, is known to promote bone growth and inhibit bone resorption [77,78], while proline, which is produced by the metabolism of lysine, has various physiological functions such as antioxidant and anti-inflammatory, and the deficiency of these metabolites may increase the risk of OA [79,80]. Our meta-analysis findings indicate that there were increased levels of tryptophan, lysine, and leucine detected in the blood samples of OA patients compared to healthy individuals. These amino acid alterations may serve as potential indicators for the development of OA.

Proline, phenylalanine, glutamine, dimethylglycine, citrulline, asparagine, and acetylcarnitine are AAs and their derivatives or metabolites [81]. They have antioxidant and anti-inflammatory effects, which can relieve the symptoms and pain of patients with OA, promote the proliferation and differentiation of chondrocytes, and facilitate the repair and regeneration of articular cartilage, as well as regulating the balance of bone metabolism, thus maintaining bone and joint health [68,82,83,84,85,86]. Our meta-analysis results reveal that the concentrations of proline, phenylalanine, glutamine, dimethylglycine, citrulline, asparagine, and acetylcarnitine were decreased in the blood samples of OA patients compared to healthy individuals. This may be due to the imbalanced diet and inadequate intake of these amino acids in OA patients. Additionally, altered metabolic states could also contribute to these changes in amino acid metabolism. For example, acetylcarnitine is produced through fatty acid metabolism, and its decreased concentration may be associated with abnormal fatty acid metabolism. Furthermore, our meta-analysis found that the blood creatinine levels were decreased in OA patients compared to healthy individuals. Creatinine is a metabolite produced by muscle metabolism [87]. Studies have linked decreased muscle mass to an increased risk of OA [88], and OA can limit physical activity, leading to decreased muscle mass [89]. Consequently, the altered concentrations of these metabolites may serve as reflections of the occurrence of OA.

Following a qualitative analysis, it can be observed that in addition to amino acid metabolism, changes also occur in glucose, lipid, and lactate metabolism. The main metabolic pathways in mitochondria include the tricarboxylic acid (TCA) cycle and oxidative phosphorylation [90]. Mitochondria play a crucial role in maintaining homeostasis in chondrotes [91]. However, in OA joints, the mitochondrial function in chondrocytes is severely disrupted, leading to increased inflammation and apoptosis, decreased autophagy, reduced mitochondrial biogenesis, and increased atabolic activity. Regulating glucose transport and glycolytic pathways in chondrocytes from OA cartilage is believed to impact the pathogenesis of the disease [56]. Lipids, with their diverse types and complex structures, play essential roles in cellular functions [92]. To respond quickly and effectively to cellular stress, all organisms require complex regulatory networks to maintain lipid and membrane balance [93]. A study has shown that for every unit increase in triglycerides, the clinical risk for knee osteoarthritis (KOA) and clinical risk for symptomatic KOA increase by nine and five, respectively [94]. In conclusion, from a metabolic perspective, glucose, lipid (fatty acids and cholesterol), and lactate metabolism are likely to be involved in the pathogenesis of OA and warrant further investigation [95,96,97].

Unfortunately, the lack of consistency in sample collection and preparation methods across multiple original studies on the metabolism of glucose, lactate, and fatty acids, among others, posed a challenge in synthesizing these findings. Variations in the measurement of outcome variables, such as metabolic concentrations, metabolite area ratios, and metabolite relative abundance spectra features, further contribute to the diverse and disparate results reported in this study. Consequently, the quantitative merging of these findings and the validation of future prospects are hindered. To address these challenges, it is imperative to establish standardized and unified protocols for studying differential metabolites between OA patients and healthy individuals.

## 5. Limitations and Perspectives

Our study still has some limitations. (1) The main problem lies in the heterogeneity of the meta-analysis results, despite the relatively consistent direction of the effect estimates. Heterogeneity is inevitable given the different populations, non-standardized protocols for biological sample collection and preparation, and the use of different analytical platforms with different coverage, sensitivity, and selectivity in different studies. (2) The names of the differential metabolites could only be extracted from the original literature, and we were unable to assess the diagnostic values of each candidate metabolic biomarker using ROC (subject operating characteristic curve) analysis. Future studies with multiple populations of osteoarthritis using standardized protocols and platforms for the quantitative detection of metabolites are needed to confirm our findings in this systematic review. (3) This study exclusively employed retrospective case–control designed studies. The inherent nature of retrospective studies poses challenges in establishing the precise temporal sequence between exposure factors and disease occurrence, limiting the ability to determine causality. Consequently, the findings of this study are limited to associations rather than causal relationships. (4) We undertook no search of the gray literature and clinical trial registry databases, which may have led to a lack of comprehensiveness in the research.

## 6. Conclusions

The main differential metabolites between OA and healthy individuals are AAs and their derivatives, including tryptophan, lysine, leucine, proline, phenylalanine, glutamine, dimethylglycine, citrulline, asparagine, acetylcarnitine and creatinine (muscle metabolic products), which could be potential biomarkers for predicting OA. However, these findings need to be further investigated and validated in larger prospective cohort studies.

## Figures and Tables

**Figure 1 nutrients-15-04191-f001:**
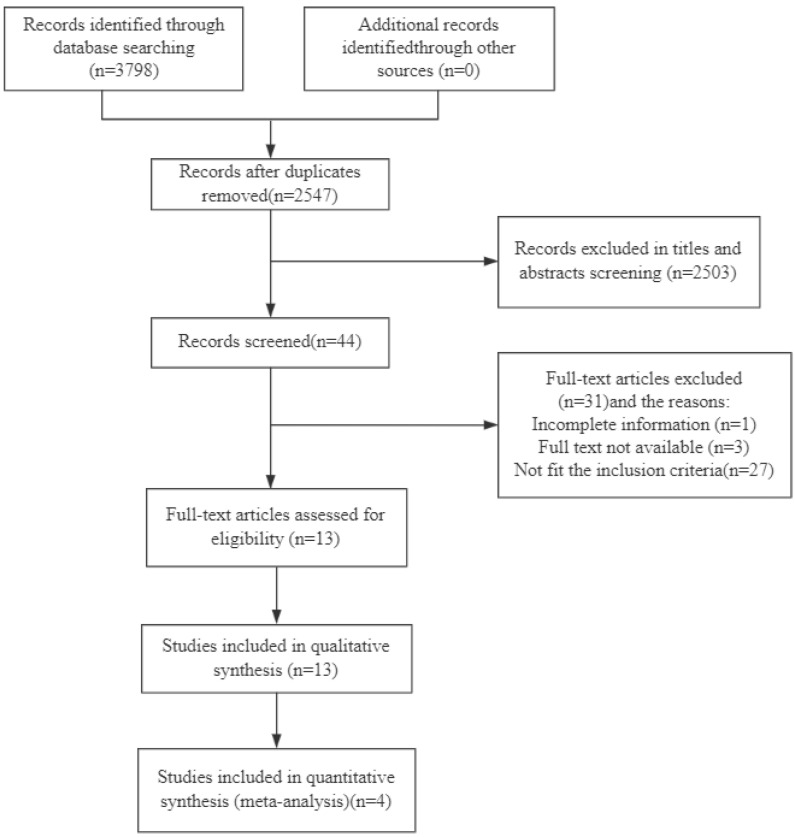
Flow diagram for the assessment of studies identified in the systematic review.

**Figure 2 nutrients-15-04191-f002:**
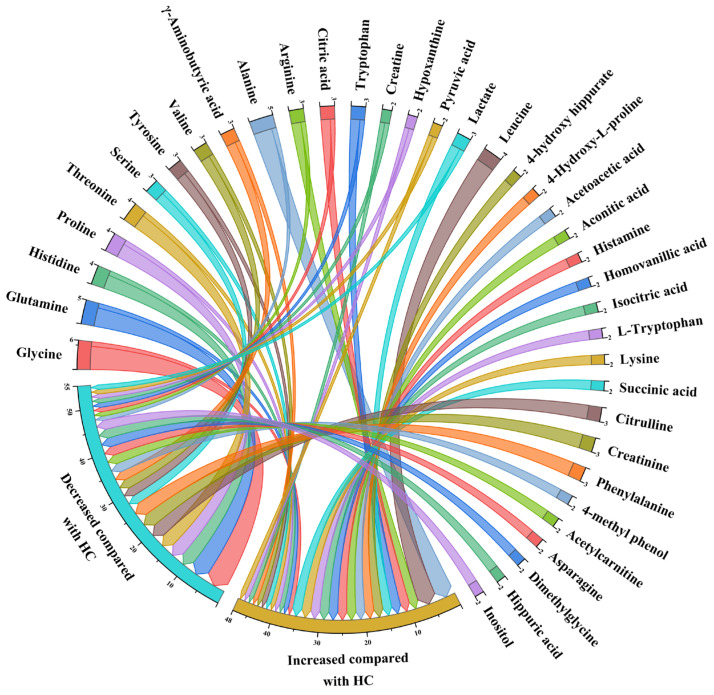
The frequency of 37 differential small molecule metabolites between OA and HC.

**Figure 3 nutrients-15-04191-f003:**
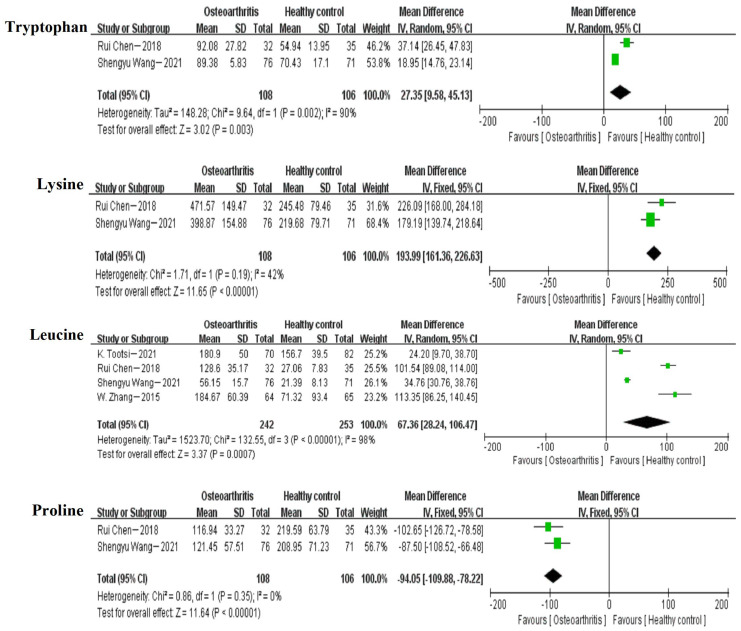
Forest plots for differential small molecule metabolites (μmol/L) between OA and HC. Data from references [38,44,47,48]. Note: The small green squares represent the point estimates for each study, and the black diamond blocks represent the combined values of the study results.

**Table 1 nutrients-15-04191-t001:** The characteristics and NOS rate of the included studies.

Author (Year)	Country	Sample	Sample Size (*n* = M/F)		BMI		Age (OA/HC)	Metabolomics Technique	NOS
			OA	HC	OA	HC			
Clara Pertusa et al. (2022) [46]	Spain	Blood serum	53	23	27.66 ± 5.53	28.04 ± 4.83	76.00 ± 9.49/70.02 ± 7.13	NMR	7
Shengyu Wang et al. (2021) [38]	China	Blood serum	40/36	35/36	NG	NG	70.5 (50–75)/61.0 (50–75)	UPLC-TQ-MS	7
Tao Kuang et al. (2021) [42]	China	Urine	10/15	10/15	NG	NG	42–70 (M); 40–66 (F)/40–60 (M); 41–68 (F)	GC-MS	5
Salah Abdelrazig et al. (2021) [43]	UK	Urine	26/48	30/38	30.23	28.34	68 (50–91)/68 (52–88)	LC-HRMS	8
K. Tootsi et al. (2021) [47]	Estonia	Blood serum	36/34	38/44	27.8 ± 3.2	26.0 ± 3.5	62 ±7/61 ± 8	LC- and FIA-MS (AbsoluteIDQ™ p180)	8
Yi Zhang (2020) [40]	China	Blood serum	14/16	15/15	NG	NG	63.45 ± 8.53/67.86 ± 7.45	GC-MS	7
Rui Chen et al. (2018) [48]	China	Blood serum	3 15/17	17/18	25.3 ± 2.3	24.3 ± 3.0	56.4 ± 3.6/54.8 ± 4.5	UPLC-TQ-MS	7
Qin Shao et al. (2017) [39]	China	Blood serum	31/69	7/13	23.4 ± 5.8	22.6 ± 2.7	62.7 ± 8.6/61.9 ± 4.1	1H-NMR	7
Kaidi Zheng et al. (2016) [45]	China	knee synovial fluid	49	21	NG	NG	NG	GC-TOF/MS; LC/MS	7
Qingmeng Zhang et al. (2015) [49]	China	Blood serum	20/20	10/10	28.2 ± 3.4	24.2 ± 2.3	58.1 ± 6.93/56.3 ± 7. 9	UPLC-MS	7
W. Zhang et al. (2016) [44]	Canada	Blood serum	64	45	33.9 ± 7.3	30.09 ± 6.7	65 ± 7/46 ± 8	UPLC	6
Xin Li et al. (2010) [50]	China	Urine	7/30	11/26	25.1 ± 3.2	23.8 ± 1.9	56.8 ± 7.2/56.3 ± 7.9	GC–MS	8
Songbin Yang et al. (2009) [41]	China	Urine	7/30	11/26	25.13 ± 3.25	24.06 ± 2.11	56.8 ± 7.1/56.0 ± 7.4	GC-MS	7

Note: M, male; F, female; NG, not given. Abbreviations: NMR, Nuclear Magnetic Resonance spectroscopy; UPLC-TQ-MS, Ultra Performance Liquid Chromatography–Time of Flight–Mass Spectrometry; GC-MS, Gas Chromatography–Flight Mass Spectrometry; LC-HRMS, Liquid Chromatography–high-Resolution Mass Spectrometry; 1H-NMR, Hydro-Nuclear Magnetic Resonance Spectrometer; GC-TOF-MS, Gas Chromatography–Time of Flight–Mass Spectrometry; LC-MS/MS, Liquid Chromatography–Electrospray Tandem Mass Spectrometry; UPLC, Ultra-Performance Liquid Chromatography; GC-MS, Gas Chromatography–Flight Mass Spectrometry.

**Table 2 nutrients-15-04191-t002:** Qualitative synthesis results of differential small molecule metabolites between OA and HC.

Trend	Differential Small Molecule Metabolites Name			
	Blood Samples		Urine Samples	Knee Synovial Fluid
Upward	**Leucine, 4-Hydroxy-L-proline, L-Tryptophan, Lysine, Succinic acid**, 2-Ketoisopropionic acid, 3-Carboxy-4-methyl-2-oxopentanoate, Acetic acid, Acetone, ADMA, fatty acidsC16: 0, fatty acidsC18: 0, fatty acidsC18: 2, fatty acidsC18: 4, fatty acidsC20: 3, fatty acidsC20: 4, fatty acidsC22: 5, fatty acidsC22: 6, fatty acid, Galactose, Glucose, Glycerol, Homocysteine, Isoleucine, Lipid, LysoPC a C16:0, LysoPC a C18:0, Methionine, Methyl-hippuric acid, N-Acetylgalactosamine, Pyruvate, Ribotide, Spermidine		**4-hydroxy hippurate, Acetoacetic acid, Aconitic acid, Histamine, Homovanillic acid, Isocitric acid, Succinic acid**, 2,3-Diaminopropionic acid, 2-Keto-glutaramic acid, 3-Nitrotyrosine, 4-Methyleneproline, 4-Methylproline, Acetylphosphate, D-glucose, Fumarate, Mannitol, Phosphoric acid, Prolyl-Glutamate, Ribose, S-Lactoylglutathione, Suberic acid, Urate	1,5-Anhydroglucitol, 8-Aminocaprylic acid, Gluconic, lactone, Tyramine
Downward	**Citrulline, Creatinine, Phenylalanine, Acetylcarnitine, Asparagine, Dimethylglycine, Inositol**, 2-aminobutyrate, 4-aminobutyrate, 4-Oxoproline, Acetate, Aminomalonic acid, Dimethylamine, fatty acidsC18: 1, fatty acidsC20: 2, glutamic acid, L-Glycine, L-Histidine, LysoPC(18:0), LysoPC a C28:1, N(CH3)3, Ornithine, Hydroxyproline, PC aa C36:6, PC ae C36:2, PC ae C38:0, Phosphocholine, Propionyl-L-carnitine, Pyridoxine, SM (OH) C14:1, SM (OH) C22:1, SM (OH) C22:2, SM (OH) C24:1, SM C16:0, SM C16:1, SM C24:0, Sphingomyelin (d18:1/16:0), Taurine, α-Aminobutyric acid, α-glucose, β-glucose		**Creatinine, 4-methyl phenol, Hippuric acid, Inositol**, 2-Hydroxyhippuric acid, 3-Methoxyphenylacetic acid, 3-Methylcrotonylglycine, 3-Oxoalanine, 4-Hydroxybutyric acid, Aminoadipic acid, Cytosine, Homocysteine sulphinic acid, Hydroxykynurenine, L-Homoserine, N-Acetyl-L-glutamate 5-semialdehyde, N-Phenylacetyl-L-glutamine, Pipecolic acid, Tartaric acid	
Inconsistent	Upward	Glycine, Histidine, Serine, Tyrosine, Valine, γ-Aminobutyric acid, **Alanine, Arginine, Tryptophan**, Creatine, Hypoxanthine, Pyruvic acid, Lactate	Proline, **Citric acid**	Glutamine, Threonine
	Downward	**Glycine, Glutamine**, Histidine, **Proline, Threonine**, **Serine**, **Tyrosine, Valine, γ-Aminobutyric acid**, Alanine, Arginine, Creatine, Hypoxanthine, Pyruvic acid, Lactate	**Glycine, Glutamine, Histidine**, Threonine, Citric acid, Tryptophan	

Note: In bold are metabolites that were studied with a frequency of 2 or more in 13 studies.

**Table 3 nutrients-15-04191-t003:** Results of meta-analysis for differential metabolites between OA and HC.

Metabolite Name	Studies for Synthesis	MD (μmol/L)	95%CI
Tryptophan	2 (1, 2)	27.35	[9.58,45.13]
Lysine	2 (1, 2)	193.99	[161.36,226.63]
Leucine	4 (1, 2, 3, 4)	67.36	[28.24,106.47]
Proline	2 (1, 2)	−94.05	[−109.88, −78.22]
Phenylalanine	2 (1, 2)	−47.12	[−52.55, −41.70]
Glutamine	2 (1, 2)	−115.66	[−230.27, −1.04]
Dimethylglycine	2 (1, 2)	−3.49	[−4.49, −2.49]
Creatinine	2 (1, 3)	−38.68	[−58.31, −19.05]
Citrulline	3 (1, 2, 3)	−21.70	[−29.25, −14.16]
Asparagine	2 (1, 2)	−28.96	[−28.05, −16.79]
Acetylcarnitine	2 (1, 2)	−6.46	[−7.41, −5.51]

Note: 1, Rui Chen et al., 2018 [48]; 2, Shengyu Wang et al., 2021 [38]; 3, W. Zhang et al., 2016 [44]; 4, K. Tootsi et al., 2021 [47].

## Data Availability

The data underlying this article are available within the article and its online Appendix A.

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
