# Peer review of "Differential Metabolites in Osteoarthritis: A Systematic Review and Meta-Analysis"

_nutrients, 2023, doi:10.3390/nu15194191_

Round 1
Reviewer 1 Report
Dear authors, you have identified almost 3800 publications that could have osteoarthritis and metabolites as a topic, of which you have read 44 in full text and have been able to include 13 in the qualitative analysis and 4 in the quantitative analysis. These studies were published in English or Chinese.
132 metabolites were included. Tryptophan, lysine, and leucine were increased; proline, phenylalanine, glutamine, dimethylglycine, creatinine, citrulline, asparagine and acetylcarnitine were decreased. For some metabolites, there are conflicting data. Possible explanations for these 11 metabolites were discussed. The limitations of the methodology, meta-analysis from cross-sectional study was named. The work provides a basis to select promising candidates from the large number for future prospective studies.
The study cannot methodologically meet the claim of finding prognostic markers for OA. Furthermore, due to the design of the selected studies, it is not possible to differentiate between pathogenesis and disease outcome.
Some discussion points should be considered differently. The authors write in line 297: Patients with OA have reduced muscle mass due to age, which leads to lower levels of creatinine anhydride. This is not plausible in age-matched controls. Rather, it is proven that reduced muscle mass predisposes to osteoarthritis (pathogenetic) and that osteoarthritis restricts movement and thus reduces muscle mass (disease consequence).
The authors also suggest that metabolite disturbances can be explained as a cause of the disease, and they find a discrepancy between attributed and observed values, e.g., for tryptophan.
It should also be considered here that osteoarthritis is a regulated physiological adaptation process of the organism to external mechanical influences, and thus chondroprotective factors may also be elevated in people with osteoarthritis.
In addition, localized inflammation via abrasion/damage proteins and TLR is physiological and useful to clean the joint surfaces.
Another aspect is the higher risk for OA in obesity and the concomitant activation of proinflammatory cascades in obese individuals. Unfortunately, weight is not addressed, is there no data on this?
OA is diagnosed more frequently in women than in men; the paper does not address this. Is there calculated data on this?
The studies were conducted in different cultures, including China and Europe with different dietary habits. Are there any explainable differences regarding the conflicting data as a result?
Formally, it is not necessary to mention the results in the text (line 219 following) if they are also presented as a table (Figure 3).
There are time jumps from past and future in the methodology, which should be unified.
Some sentences are not complete, verbs are missing, ...
Author Response
Response to Reviewer 1 Comments
Dear reviewer, thank you very much for reviewing our article seriously and rigorously and giving very valuable suggestions, we have seriously revised it according to your suggestions one by one and marked the revised version of the manuscript with red color to make it convenient for you to review it again, and thank you once again for your hard work on our manuscript.
Point 1:The study cannot methodologically meet the claim of finding prognostic markers for OA. Furthermore, due to the design of the selected studies, it is not possible to differentiate between pathogenesis and disease outcome.
Response 1:
Thank you very much for your valuable suggestions. Our study consisted of case-control studies only, and due to its inherent nature, it is indeed impossible to differentiate between pathogenesis and disease outcome, which we have taken into account and written in lines 336-340 in the main text. However, based on your valuable comments, we can improve the presentation of the study objectives.
The aim of our study is to screen promising candidates from a large number of samples to provide a basis for future prospective studies, and ultimately to screen biomarkers with predictive value for the development of OA. In other words, before a patient with OA develops physical symptoms, or before OA is diagnosed by imaging, we can assess the risk of OA by testing blood, urine, or synovial fluid samples from subjects and observing changes in certain metabolites in the samples, and detecting OA early enough to allow early intervention with a view to improving the patient's quality of life and reducing the healthcare burden. This was our wish, but after your reminder, in the case of this study, the claim of finding predictive markers for OA was indeed not met due to the limitations of the study design.
Therefore, we decided to modify our elaboration. One, to revise the background section of the abstract to correct the statement about predicting the disease and understanding its pathogenesis; and two, to revise lines 96-97 by deleting the phrase "identifying biomarkers for possible future predictive validation" and revising it to read "will provide a useful basis for future metabolomics studies on OA samples".
Point 2: Some discussion points should be considered differently. The authors write in line 297: Patients with OA have reduced muscle mass due to age, which leads to lower levels of creatinine anhydride. This is not plausible in age-matched controls. Rather, it is proven that reduced muscle mass predisposes to osteoarthritis (pathogenetic) and that osteoarthritis restricts movement and thus reduces muscle mass (disease consequence).
Response 2: Thank you very much for providing us with your professional advice. We have made the necessary corrections to our statement based on your suggestions, as shown in lines 317-319.
Point 3: The authors also suggest that metabolite disturbances can be explained as a cause of the disease, and they find a discrepancy between attributed and observed values, e.g., for tryptophan. It should also be considered here that osteoarthritis is a regulated physiological adaptation process of the organism to external mechanical influences, and thus chondroprotective factors may also be elevated in people with osteoarthritis.In addition, localized inflammation via abrasion/damage proteins and TLR is physiological and useful to clean the joint surfaces.
Response 3:Thank you very much for providing us with your valuable and professional advice, which has helped us gain a better understanding of the between them and how they relate back to our main theme. Based on your suggestions, we have made the necessary corrections to our exposition, as shown in lines 299-302.
Point 4: Another aspect is the higher risk for OA in obesity and the concomitant activation of proinflammatory cascades in obese individuals. Unfortunately, weight is not addressed, is there no data on this?
Response 4:Thank you very much for your valuable comments, the issue of weight is really important, most of the included original studies provided BMI, according to your suggestion we have added the BMI of each original study in Table 1.
Point 5: OA is diagnosed more frequently in women than in men; the paper does not address this. Is there calculated data on this?
Response 5: The original studies we included provided the sample sizes of the studies they included, and in terms of the number of OA samples included in each of the original studies, most of the studies included more females than males. Again, we have added this raw data to Table 1.
Point 6: The studies were conducted in different cultures, including China and Europe with different dietary habits. Are there any explainable differences regarding the conflicting data as a result?
Response 6:Thank you very much for helping us think about this issue in many ways. We carefully reviewed the extracted data and did not find differences with opposite trends Small molecule metabolites were significantly associated with different cultural backgrounds. For example, the trend for glycine was not consistent, with three studies reporting a decreasing trend in osteoarthritis patients compared with healthy individuals, two of which were in the geographic region of China and one in Spain, and one study reporting a decreasing trend in osteoarthritis patients compared with healthy individuals, with the geographic region of the study being China. The situation was similar for many other differential metabolites, with the same geographic area of study but inconsistent trends. We have also considered the possibility that sample differences (blood, urine, synovial fluid) may also contribute to trend inconsistencies, but the number of original studies in the field is currently insufficient to be able to compare to analyze these differential small molecule metabolites with inconsistent trends. Therefore, unfortunately, as much as we would like to interpret this result, the amount of data is not sufficient to draw reliable conclusions.
Point 7: Formally, it is not necessary to mention the results in the text (line 219 following) if they are also presented as a table (Figure 3).
Response 7: Thank you very much for your valuable suggestion, it is indeed possible to read the results clearly by looking directly at the graphs, no don't have to mention it again in the text, so we have deleted this part of the content.
Point 8: There are time jumps from past and future in the methodology, which should be unified.
Response 8: Thank you for your patience in pointing out our mistakes. After another careful reading, we have found 2 errors in the tenses of the Methods section. One is that line 115 should use past tense, which we have already corrected. Second, line 128 'The studies are evaluated by a "star system" based on three aspects......' should be changed to 'The studies were.......' , which we have also corrected.
Point 9:Some sentences are not complete, verbs are missing, ...
Response 9: Thank you very much for your careful review of our manuscript and your valuable suggestions. We are very sorry, because our native language is not English, our limited English level has read it many times, but we still can't find the sentences with missing verbs, can you please point them out for us, thank you very much for your hard work on our manuscript.

Reviewer 2 Report
The authors present a systematic review and meta-analysis to identify differential small molecule metabolites between osteoarthritis (OA) patients and healthy individuals.
The manuscript is well written and organized. The topic is timely, novel and would be interesting to readers. The results are important, as they provide information to identify biomarkers that may have potential for future predictive validation.
The authors carry out a systematic review and meta-analysis, methodologically correct. They indicate it was reported by protocol PRISMA-P and registered in PROSPERO.
The authors detail the selection criteria, the Search strategy, the study selection, the risk of bias assessment, the data collection and the data shyntesis.
the results are clearly displayed and facilitate the Flow diagram for the assessment of studies identified in the systematic review.
The authors are aware of the limitations of the article, based on the heterogeneity found in the selected articles.
Minor revisions:
- There are a few areas that could be improved or clarified, detailed below:
The authors detail in rows 152 and 153 (data synthesis) “Funnel plots and the Egger’s test were used to assess publication bias when feasible (10 or more studies)”, but finally they are not included in the results, I suppose because the number of articles analyzed was less. That is why they should clarify it or, where appropriate, remove it from said section.
- There is a grammatical error in row 98, instead of PROPSERO it should say PROSPERO that needs to be corrected.
Author Response
Response to Reviewer 2 Comments
Dear reviewer, thank you very much for your affirmation of our manuscript and your valuable suggestions for revision. I have explained your suggestions one by one and marked them in orange in the revised version of the manuscript for your convenience. Thank you again for your contribution.
Point 1: The authors detail in rows 152 and 153 (data synthesis) “Funnel plots and the Egger’s test were used to assess publication bias when feasible (10 or more studies)”, but finally they are not included in the results, I suppose because the number of articles analyzed was less. That is why they should clarify it or, where appropriate, remove it from said section.
Response 1: Yes, we were unable to assess publication bias using funnel plots and Egger's test because the number of included literatures were all less than 10. I have added this sentence to the "Meta-analysis" section of the results.
Point 2:There is a grammatical error in row 98, instead of PROPSERO it should say PROSPERO that needs to be corrected.
Response 2:In line 101, it has been changed to "PROSPERO".

Reviewer 3 Report
The authors of the article have performed a meta-analysis of the metabolites involved in the process of osteoarthritis (OA). The review has a clear methodology and shows that a variety of amino acids (e.g. tryptophan, lysine, proline) are differentially regulated in OA.
This review provides a useful basis for future metabolomics investigation for OA samples. The authors should address the following points,
1. What is the meaning of the abbreviation of HC ? Please clarify in the text.
2. Are there any differences between the studies using blood, urine and synovial fluid samples ? Furthermore, it is unclear from the table 1 what samples are examined in each study.
3. Apart from amino acids in these studies, is there evidence of changes in glucose, lactate fatty acid metabolism in these studies ? Previous investigations have postulated that OA tissues undergo oxidative phosphorylation compared to glycolysis in normal cartilage.
4. What are the pathway changes in OA resulting from the changes in amino acid metabolism ? Discussion about the pathways involved and the differences in normal cartilage should be stated.
5. A figure summarizing the changes in metabolites between normal/healthy cartilage and osteoarthritic tissues should be included in a revised manuscript to help readers.
Author Response
Response to Reviewer 3 Comments
Dear reviewer, thank you very much for your careful and rigorous review of our manuscript, and thank you for giving us very valuable advice, we seriously according to each of your suggestions one by one to make changes and use the blue color in the text of the mark, so that it is convenient for you to review again.
Point 1: What is the meaning of the abbreviation of HC ? Please clarify in the text.
Response 1: HC means “healthy controls”, as clarified in line 28 of the abstract and line 106 of the main text, respectively.
Point 2: Are there any differences between the studies using blood, urine and synovial fluid samples ? Furthermore, it is unclear from the table 1 what samples are examined in each study.
Response 2: From the results of qualitative analysis, some metabolites showed different trends in different samples, for example tryptophan was increasing in blood samples and decreasing in urine samples; glutamine and threonine were decreasing in both blood and urine samples but increasing in knee synovial fluid; proline was decreasing in blood samples and increasing in urine samples was increased. Thus, there are differences between the samples studied. It is with this in mind that we have used data from the same samples in our quantitative synthesis. We have clarified this point in lines 182-183 of the main text. Also thank you very much for the valuable suggestion that we have added 1 column to Table 1 on line 185 to indicate which samples were examined in each study.
Point 3: Apart from amino acids in these studies, is there evidence of changes in glucose, lactate fatty acid metabolism in these studies ? Previous investigations have postulated that OA tissues undergo oxidative phosphorylation compared to glycolysis in normal cartilage.
Response 3:It is true that in these studies changes in glucose, lactate, and fatty acid metabolism in addition to amino acids have been shown. In Table 2, which was added in response to your fifth suggestion, it can be seen that glucose and fatty acid metabolism are elevated in blood samples from osteoarthritis patients compared to healthy individuals, and that the trend in lactate metabolism is inconsistent. Unfortunately there are too few articles examining the differences in glucose, lactate, and fatty acid metabolism between healthy individuals and patients with osteoarthritis, and no specific concentration values are given to allow for quantitative combining and thus Meta-analysis. We found this out during our research and discussed it in the last paragraph of the 4. Discussion section, but after your reminder we realized that we were not specific enough, so we decided to elaborate on this point in more detail to help the reader understand it. It was also brought to our attention that it would be useful to discuss the results of the qualitative analysis in addition to the results of the quantitative analysis, which is supplemented by lines 321-339.
Point 4: What are the pathway changes in OA resulting from the changes in amino acid metabolism ? Discussion about the pathways involved and the differences in normal cartilage should be stated.
Response 4:Thank you for your invaluable professional insights, which have greatly assisted us in our efforts to conduct thorough research. We have taken your suggestions into careful consideration and made appropriate additions to our work, which can be found in lines 265-276.
Point 5: A figure summarizing the changes in metabolites between normal/healthy cartilage and osteoarthritic tissues should be included in a revised manuscript to help readers.
Response 5:
Thank you very much for your valuable suggestions. Taking into account your second suggestion, we decided to summarize the changes in metabolites in blood, urine and synovial fluid samples from osteoarthritis patients and healthy individuals in a table in Table 2 and add the corresponding descriptions in rows 215-221 of the qualitative synthesis section of 3.4. We hope that the tabular format will help the reader to more visually compare the changes in metabolites in different samples from osteoarthritis patients and healthy individuals.

Round 2
Reviewer 1 Report
Dear authors,
thank you for the changes made with the 2nd proposal and the kind responses.
To optimize the table 1 the sample size should be given in male/female or total (if male/female is not known)
Reviewer 3 Report
The authors have answered my questions appropriately.